# Legged Locomotion in Challenging Terrains using Egocentric Vision

**Ananye Agarwal**[* 1] **Ashish Kumar**[* 2], **Jitendra Malik**[†2], **Deepak Pathak**[†1]
[1]Carnegie Mellon University, [2]UC Berkeley

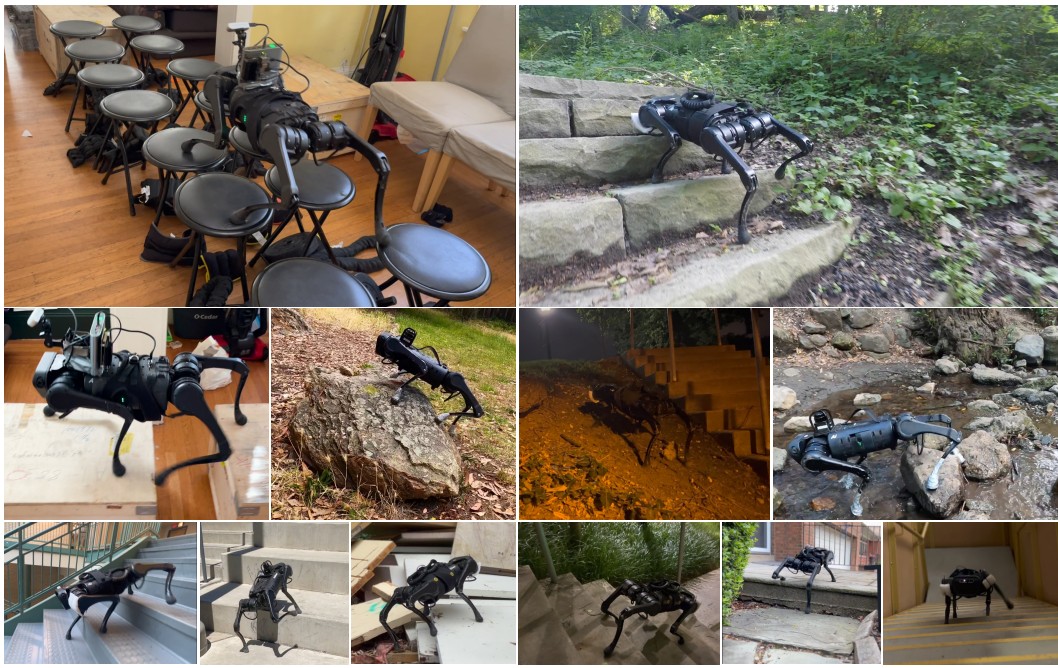

Figure 1: Our robot can traverse a variety of challenging terrain in indoor and outdoor environments, urban and natural settings during day and night using a single front-facing depth camera. The robot can traverse curbs, stairs and moderately rocky terrain. Despite being much smaller than other commonly used legged robots, it is able to climb stairs and curbs of a similar height. Videos at https://vision-locomotion.github.io

**Abstract:** Animals are capable of precise and agile locomotion using vision. Replicating this ability has been a long-standing goal in robotics. The traditional approach has been to decompose this problem into elevation mapping and foothold planning phases. The elevation mapping, however, is susceptible to failure and large noise artifacts, requires specialized hardware, and is biologically implausible. In this paper, we present the first end-to-end locomotion system capable of traversing stairs, curbs, stepping stones, and gaps. We show this result on a medium-sized quadruped robot using a single front-facing depth camera. The small size of the robot necessitates discovering specialized gait patterns not seen elsewhere. The egocentric camera requires the policy to remember past information to estimate the terrain under its hind feet. We train our policy in simulation. Training has two phases - first, we train a policy using reinforcement learning with a cheap-to-compute variant of depth image and then in phase 2 distill it into the final policy that uses depth using supervised learning. The resulting policy transfers to the real world and is able to run in real-time on the limited compute of the robot. It can traverse a large variety of terrain while being robust to perturbations like pushes, slippery surfaces, and rocky terrain. Videos are at https://vision-locomotion.github.io.

---

[*]Equal Contribution. [†]Equal Advising.

6th Conference on Robot Learning (CoRL 2022), Auckland, New Zealand.

# 1   Introduction

Of what use is vision during locomotion? Clearly, there is a role of vision in navigation – using maps or landmarks to find a trajectory in the 2D plane to a distant goal while avoiding obstacles. But given a local direction in which to move, it turns out that both humans [1] and robots [2, 3] can do remarkably well at blind walking. Where vision becomes necessary is for locomotion in challenging terrains. In an urban environment, staircases are the most obvious example. In the outdoors, we can deal with rugged terrain such as scrambling over rocks, or stepping from stone to stone to cross a stream of water. There is a fair amount of scientific work studying this human capability and showing tight coupling of motor control with vision [4, 5, 6]. In this paper, we will develop this capability for a quadrupedal walking robot equipped with egocentric depth vision. We use a reinforcement learning approach trained in simulation, which we are directly able to transfer to the real world. Figure 1 and the accompanying videos shows some examples of our robot walking guided by vision.

Humans receive an egocentric stream of vision which is used to control feet placement, typically without conscious planning. As children we acquire it through trial and error [7] but for adults it is an automatized skill. Its unconscious execution should not take away from its remarkable sophistication. The footsteps being placed now are based on information collected some time ago. Typically, we don't look at the ground underneath our feet, rather at the upcoming piece of ground in front of us a few steps away[1, 4, 5, 6]. A short term memory is being created which persists long enough to guide foot placement when we are actually over that piece of ground. Finally, note that we learn to walk through bouts of steps, not by executing pre-programmed gaits [7].

We take these observations about human walking as design principles for the visually-based walking controller for an A1 robot. The walking policy is trained by reinforcement learning with a recurrent neural network being used as a short term memory of recent egocentric views, proprioceptive states, and action history. Such a policy can maintain memory of recent visual information to retrieve characteristics of the terrain under the robot or below the rear feet, which might no longer be directly visible in the egocentric view.

In contrast, prior locomotion techniques rely on the metric elevation map of the terrain around and under the robot [8, 9, 10] to plan foot steps and joint angles. The elevation map is constructed by fusing information from multiple depth images (collected over time). This fusion of depth images into a single elevation map requires the relative pose between cameras at different times. Hence, tracking is required in the real world to obtain this relative pose using visual or inertial odometry. This is challenging because of noise introduced in sensing and odometry, and hence, previous methods add different kinds of structured noise at training time to account for the noise due to pose estimation drift [11, 12, 13]. The large amount of noise hinders the ability of such systems to perform reliably on gaps and stepping stones. We use vision as a first class citizen and show all the uneven terrain capabilities along with a high success rate on crossing gaps and stepping stones.

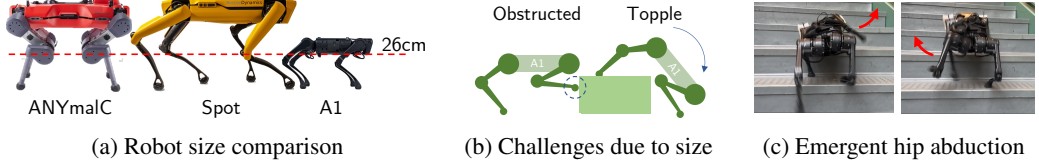

(a) Robot size comparison        (b) Challenges due to size        (c) Emergent hip abduction

Figure 2: A smaller robot (a) faces challenges in climbing stairs and curbs due to the stair obstructing its feet while going up and a tendency to topple over when coming down (b). Our robot deals with this by climbing using a large hip abduction that automatically emerges during training (c).

The design principle of not having pre-programmed gait priors turns out to be quite advantageous for our relatively small robot [1] (fig. 2). Predefined gait priors or reference motions fail to generalize to obstacles of even a reasonable height because of the relatively small size of the quadruped. The emergent behaviors for traversing complex terrains without any priors enable our robot with a hip joint height of 28cm to traverse the stairs of height upto 25cm, 89% relative to its height, which is significantly higher than any existing methods which typically rely on gait priors.

Since our robot is small and inexpensive, it has limited onboard compute and sensing. It uses a single front-facing D435 camera for exteroception. In contrast, AnymalC has four such cameras in addition to two dome lidars. Similarly, Spot has 5 depth cameras around its body. Our policy computes actions

---

[1]A1 standing height is 40cm as measured by us. Spot, ANYmalC both are 70cm tall reported here and here.

with a single feedforward pass and requires no tracking. This frees us from running optimization for MPC or localization which requires expensive hardware to run in real-time.

Overall, this use of learning "all the way" and the tight coupling of egocentric vision with motor control are the distinguishing aspects of our approach.

## 2 Method: Legged Locomotion from Egocentric Vision

Our goal is to learn a walking policy that maps proprioception and depth input to target joint angles at 50Hz. Since depth rendering slows down the simulation by an order of magnitude, directly training this system using reinforcement learning (RL) would require billions of samples to converge making this intractable with current simulations. We therefore employ a two-phase training scheme. In phase 1, we use low resolution scandots located under the robot as a proxy for depth images. Scandots refer to a set of $(x, y)$ coordinates in the robot's frame of reference at which the height of the terrain is queried and passed as observation at each time step (fig. 3). These capture terrain geometry and are cheap to compute. In phase 2, we use depth and proprioception as input to an RNN to implicitly track the terrain under the robot and directly predict the target joint angles at 50Hz. This is supervised with actions from the phase 1 policy. Since supervised learning is orders of magnitude more sample efficient than RL, our proposed pipeline enables training the whole system on a single GPU in a few days. Once trained, our deployment policy does not construct metric elevation maps, which typically rely on metric localization, and instead directly predicts joint angles from depth and proprioception.

One potential failure mode of this two-phase training is that the scandots might contain more information than what depth can infer. To get around this, we choose scandots and camera field-of-view such that phase 2 loss is low. We formally show that this guarantees that the phase 2 policy will have close to optimal performance in Thm 2.1 below.

**Theorem 2.1.** $\mathcal{M} = (\mathcal{S}, \mathcal{A}, P, R, \gamma)$ *be an MDP with state space* $\mathcal{S}$, *action space* $\mathcal{A}$, *transition function* $P : \mathcal{S} \times \mathcal{A} \to \mathcal{S}$, *reward function* $R : \mathcal{A} \times \mathcal{S} \to \mathbb{R}$ *and discount factor* $\gamma$. *Let* $V^1(s)$ *be the value function of the phase 1 policy that is trained to be close to optimal value function* $V^*(s)$, *i.e.,* $|V^*(s) - V^1(s)| < \epsilon \, \forall s \in \mathcal{S}$, *and* $\pi^1(s)$ *be the greedy phase 1 policy obtained from* $V^1(s)$. *Suppose the phase 2 policy operates in a different state space* $\mathcal{S}'$ *given by a mapping* $f : \mathcal{S} \to \mathcal{S}'$. *If the phase 2 policy is close to phase 1* $|\pi^1(s) - \pi^2(f(s))| < \eta \, \forall \, s$ *and* $R, P$ *are Lipschitz continuous, then the return of phase 2 policy is close to optimal everywhere, i.e.,* $\forall s, \left| V^*(s) - V^{\pi^2}(f(s)) \right| < \frac{2\epsilon\gamma + \eta c}{1-\gamma}$ *where* $c \propto \sum_{s \in \mathcal{S}} V^*(s)$ *is a large but bounded constant. (proof in sec. A)*

We instantiate our training scheme using two different architectures. The monolithic architecture is an RNN that maps from raw proprioception and vision data directly to joint angles. The RMA architecture follows [3, 14], and contains an MLP base policy that takes $\boldsymbol{\gamma}_t$ (which encodes the local terrain geometry) along with the extrinsics vector $\mathbf{z}_t$ (which encodes environment parameters [3]), and proprioception $\mathbf{x}_t$ to predict the target joint angles. An estimate of $\boldsymbol{\gamma}_t$ is generated by an RNN that takes proprioception and vision as inputs. While the monolithic architecture is conceptually simpler, it implicitly tracks $\boldsymbol{\gamma}_t$ and $\mathbf{z}_t$ in its weights and is hard to disentangle. In contrast, the RMA architecture allows direct access to each input ($\boldsymbol{\gamma}_t$ or $\mathbf{z}_t$) through latent vectors. This allows the possibility of swapping sensors (like replacing depth by RGB) or using one stream to supervise the other while keeping the base motor policy fixed.

### 2.1 Phase 1: Reinforcement Learning from Scandots

Given the scandots $\mathbf{m}_t$, proprioception $\mathbf{x}_t$, commanded linear and angular velocity $\mathbf{u}_t^{\mathrm{cmd}} = \left(v_x^{\mathrm{cmd}}, \omega_z^{\mathrm{cmd}}\right)$ we learn a policy using PPO without gait priors and with reward functions that minimize energetics to walk on a variety of terrains. Proprioception consists of joint angles, joint velocities, angular velocity, roll and pitch measured by onboard sensors in addition to the last policy actions $\mathbf{a}_{t-1}$. Let $\mathbf{o}_t = (\mathbf{m}_t, \mathbf{x}_t, \mathbf{u}_t^{\mathrm{cmd}})$ denote the observations. The RMA policy also takes privileged information $\mathbf{e}_t$ as input which includes center-of-mass of robot, ground friction, and motor strength.

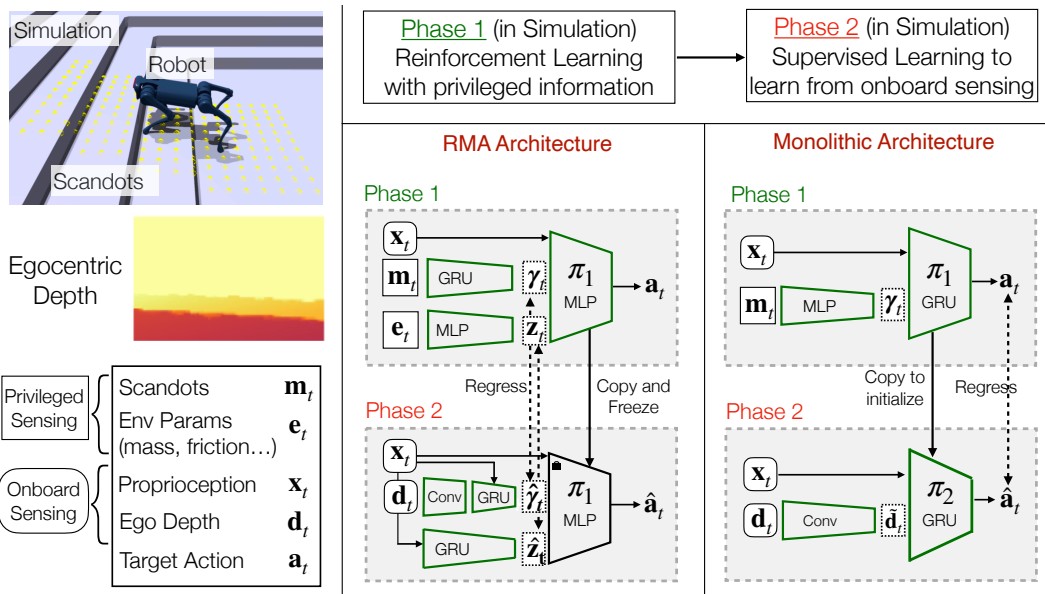

Figure 3: We train our locomotion policy in two phases to avoid rendering depth for too many samples. In phase 1, we use RL to train a policy $\pi^1$ that has access to scandots that are cheap to compute. In phase 2, we use $\pi^1$ to provide ground truth actions which another policy $\pi^2$ is trained to imitate. This student has access to depth map from the front camera. We consider two architectures (1) a monolithic one which is a GRU trained to output joint angles with raw observations as input (2) a decoupled architecture trained using RMA [3] that is trained to estimate vision and proprioception latents that condition a base feedforward walking policy.

**Monolithic** The scandots $\mathbf{m}_t$ are first compressed to $\boldsymbol{\gamma}_t$ and then passed with the rest of the observations to a GRU that predicts the joint angles.

$$\boldsymbol{\gamma}_t = \text{MLP}\left(\mathbf{m}_t\right) \tag{1}$$

$$\mathbf{a}_t = \text{GRU}_t\left(\mathbf{x}_t, \boldsymbol{\gamma}_t, \mathbf{u}_t^{\text{cmd}}\right) \tag{2}$$

the subscript $t$ on the GRU indicates that it is stateful.

**RMA** Instead of using a monolithic memory based architecture for the controller, we use an MLP as the controller, pushing the burden of maintaining memory and state on the various inputs to the MLP. Concretely, we process the environment parameters ($\mathbf{e}_t$) with an MLP and the scandots ($\mathbf{m}_t$) with a GRU to get $\mathbf{z}_t$ and $\boldsymbol{\gamma}_t$ respectively which are given as input to the base feedforward policy.

$$\boldsymbol{\gamma}_t = \text{GRU}_t\left(\mathbf{m}_t\right) \tag{3}$$

$$\mathbf{z}_t = \text{MLP}\left(\mathbf{e}_t\right) \tag{4}$$

$$\mathbf{a}_t = \text{MLP}\left(\mathbf{x}_t, \boldsymbol{\gamma}_t, \mathbf{z}_t, \mathbf{u}_t^{\text{cmd}}\right) \tag{5}$$

Both the phase 1 architectures are trained using PPO [15] with backpropagation through time [16] truncated at 24 timesteps.

**Rewards** We extend the reward functions proposed in [3, 17] to simply penalizing the energy consumption along with additional penalties to prevent damage to hardware on complex terrain (sec. B). Importantly, we do not impose any gait priors or predefined foot trajectories and let optimal gaits that are stable and natural to emerge for the task.

- *Absolute work penalty* $-|\boldsymbol{\tau} \cdot \mathbf{q}|$ where $\boldsymbol{\tau}$ are the joint torques. We use the absolute value so that the policy does not learn to get positive reward by exploiting inaccuracies in contact simulation.
- *Command tracking* $v_x^{\text{cmd}} - |v_x^{\text{cmd}} - v_x| - |\omega_z^{\text{cmd}} - \omega_z|$ where $v_x$ is velocity of robot in forward direction and $\omega_z$ is yaw angular velocity ($x, z$ are coordinate axes fixed to the robot).
- *Foot jerk penalty* $\sum_{i \in \mathcal{F}} \|\mathbf{f}_t^i - \mathbf{f}_{t-1}^i\|$ where $\mathbf{f}_t^i$ is the force at time $t$ on the $i^{\text{th}}$ rigid body and $\mathcal{F}$ is the set of feet indices. This prevents large motor backlash.

- *Feet drag penalty* $\sum_{i \in \mathcal{F}} \mathbb{I}\left[f_z^i \geq 1\mathrm{N}\right] \cdot \left(\left|v_x^i\right| + \left|v_y^i\right|\right)$ where $\mathbb{I}$ is the indicator function, and $v_x^i, v_y^i$ is velocity of $i^{\mathrm{th}}$ rigid body. This penalizes velocity of feet in the horizontal plane if in contact with the ground preventing feet dragging on the ground which can damage them.
- *Collision penalty* $\sum_{i \in \mathcal{C} \cup \mathcal{T}} \mathbb{I}\left[\mathbf{f}^i \geq 0.1\mathrm{N}\right]$ where $\mathcal{C}, \mathcal{T}$ are the set of calf and thigh indices. This penalizes contacts at the thighs and calves of the robot which would otherwise graze against edges of stairs and discrete obstacles.
- *Survival bonus* constant value 1 at each time step to prioritize survival over following commands in challenging situations.

**Training environment** Similar to [18] we generate different sets of terrain (fig. 5) of varying difficulty level. Following [3], we generate fractal variations over each of the terrains to get robust walking behaviour. At training time, the environments are arranged in a $6 \times 10$ matrix with each row having terrain of the same type and difficulty increasing from left to right. We train with a curriculum over terrain [18] where robots are first initialized on easy terrain and promoted to harder terrain if they traverse more than half its length. They are demoted to easier terrain if they fail to travel at least half the commanded distance $v_x^{\mathrm{cmd}} T$ where $T$ is maximum episode length. We randomize parameters of the simulation (tab. 3) and add small i.i.d. gaussian noise to observations for robustness (tab. 2).

## 2.2 Phase 2: Supervised Learning

In phase 2, we use supervised learning to distil the phase 1 policy into an architecture that only has access to sensing available onboard: proprioception ($\mathbf{x}_t$) and depth $\mathbf{d}_t$.

**Monolithic** We create a copy of the recurrent base policy 2. We preprocess the depth map through a convnet before passing it to the base policy.

$$\tilde{\mathbf{d}}_t = \mathrm{ConvNet}\left(\mathbf{d}_t\right) \tag{6}$$

$$\hat{\mathbf{a}}_t = \mathrm{GRU}_t(\mathbf{x}_t, \tilde{\mathbf{d}}_t, \mathbf{a}_t^{\mathrm{cmd}}) \tag{7}$$

We train with DAgger [19] with truncated backpropagation through time (BPTT) to minimize mean squared error between predicted and ground truth actions $\|\hat{\mathbf{a}}_t - \mathbf{a}_t\|^2$. In particular, we unroll the student inside the simulator for $N = 24$ timesteps and then label each of the states encountered with the ground truth action $\mathbf{a}_t$ from phase 1.

**RMA** Instead of retraining the whole controller, we only train estimators of $\boldsymbol{\gamma}_t$ and $\mathbf{z}_t$, and use the same base policy trained in phase 1 (eqn. 5). The latent $\hat{\boldsymbol{\gamma}}$, which encodes terrain geometry, is estimated from history of depth and proprioception using a GRU. Since the camera looks in front of the robot, proprioception combined with depth enables the GRU to implicitly track and estimate the terrain under the robot. Similar to [3], history of proprioception is used to estimate extrinsics $\hat{\mathbf{z}}$.

$$\tilde{\mathbf{d}}_t = \mathrm{ConvNet}\left(\mathbf{d}_t\right) \tag{8}$$

$$\hat{\boldsymbol{\gamma}}_t = \mathrm{GRU}_t\left(\mathbf{x}_t, \mathbf{u}_t^{\mathrm{cmd}}, \tilde{\mathbf{d}}_t\right) \tag{9}$$

$$\hat{\mathbf{z}}_t = \mathrm{GRU}_t\left(\mathbf{x}_t, \mathbf{u}_t^{\mathrm{cmd}}\right) \tag{10}$$

$$\mathbf{a}_t = \mathrm{MLP}\left(\mathbf{x}_t, \mathbf{u}_t^{\mathrm{cmd}}, \hat{\boldsymbol{\gamma}}_t, \hat{\mathbf{z}}_t\right) \tag{11}$$

As before, this is trained using DAgger with BPTT. The vision GRU 9 and convnet 8 are jointly trained to minimize $\|\hat{\boldsymbol{\gamma}}_t - \boldsymbol{\gamma}_t\|^2$ while the proprioception GRU 10 minimizes $\|\hat{\mathbf{z}}_t - \mathbf{z}_t\|^2$.

**Deployment** The student can be deployed as-is on the hardware using only the available onboard compute. It is able to handle camera failures and the asynchronous nature of depth due to the randomizations we apply during phase 1. It is robust to pushes, slippery surfaces and large rocky surfaces and can climb stairs, curbs, and cross gaps and stepping stones.

## 3 Experimental Setup

We use the Unitree A1 robot pictured in Fig. 2. The robot has 12 actuated joints. The robot has a front-facing Intel RealSense depth camera in its head. The onboard compute consists of the UPboard and a Jetson NX. The policy operates at 50Hz and sends joint position commands which are converted to

| Terrain | Average $x$-Displacement ($\uparrow$) | | | | Mean Time to Fall (s) | | | |
| --- | --- | --- | --- | --- | --- | --- | --- | --- |
| | RMA | MLith | Noisy | Blind | RMA | MLith | Noisy | Blind |
| Slopes | 43.98 | 44.09 | 36.14 | 34.72 | 88.99 | 85.68 | 70.25 | 67.07 |
| Stepping Stones | 18.83 | 20.72 | 1.09 | 1.02 | 34.3 | 41.32 | 2.51 | 2.49 |
| Stairs | 31.24 | 42.4 | 6.74 | 16.64 | 69.99 | 90.48 | 15.77 | 39.17 |
| Discrete Obstacles | 40.13 | 28.64 | 29.08 | 32.41 | 85.17 | 57.53 | 59.3 | 66.33 |
| Total | 134.18 | 135.85 | 73.05 | 84.79 | 278.45 | 275.01 | 147.83 | 175.06 |

Table 1: We measure the average displacement along the forward axis and mean time to fall for all methods on different terrains in simulation. For each method, we train a single policy for all terrains and use that for evaluation. We see that the monolithic (MLith) and RMA architectures of our method outperform the noisy and blind baselines by 60-90% in terms of total mean time to fall and average displacement. Vision is not strictly necessary for traversing slopes and the baselines make significant progress on this terrain, however, MLith and RMA travel upto 25% farther. The difference is more stark on stepping stones where blind and noisy baselines barely make any progress due to not being able to locate positions of the stones, while MLith and RMA travel for around 20m. Noisy and blind make some progress on stairs and discrete obstacles, but our methods travel upto 6.3 times farther.

torques by a low-level PD controller running at $400$Hz. Depth map is obtained from a Intel RealSense camera inside the head of the robot. The camera captures images every $100$ms $\pm 20$ms at a resolution of $480 \times 848$. We preprocess the image by cropping 200 white pixels from the left, applying nearest neighbor hole-filling and downsampling to $58 \times 87$. This is passed through a backbone to obtain the compressed $\tilde{\mathbf{d}}_t$ 8, 6 which is sent over a UDP socket to the base policy. This has a latency of $10 \pm 10$ms which we account for during phase 2.

We use the IsaacGym (IG) simulator with the legged_gym library [18] to train our walking policies. We construct a large terrain map with 100 sub-terrains arranged in a $20 \times 10$ grid. Each row has the same type of terrain arranged in increasing difficulty while different rows have different terrain.

**Baselines** We compare against two baselines, each of which uses the same number of learning samples for both RL phase and supervised learning phase.

- **Blind policy** trained with the scandots observations $\mathbf{m}_t$ masked with zeros. This baseline must rely on proprioception to traverse terrain and helps quantify the benefit of vision for walking.
- **Noisy** Methods which rely on elevation maps need to fuse multiple depth images captured over time to obtain a complete picture of terrain under and around the robot. This requires camera pose relative to the first depth input, which is typically estimated using vision or inertial odometry [11, 13, 12]. However, these pose estimates are typically noisy resulting in noisy elevation maps [8]. To handle this, downstream controllers trained on this typically add a large noise in the elevation maps during training. Similar to [8], we train a teacher with ground truth, noiseless elevation maps in phase 1 and distill it to a student with large noise, with noise model from [8], added to the elevation map. We simulate a latency of 40ms in both the phases of training to match the hardware. This baseline helps in understanding the effect on performance when relying on pose estimates which introduce additional noise in the pipeline.

## 4 Results and Analysis

**Simulation Results** We report mean time to fall and mean distance travelled before crashing for different terrain and baselines in Table 1. For each method, we train a single policy for all terrains and use that for evaluation. Although the blind policy makes non trivial progress on stairs, discrete obstacles and slopes, it is significantly less efficient at traversing these terrains. On slopes our methods travel upto 27% farther implying that the blind baseline crashes early. Similarly, on stairs and discrete obstacles the distance travelled by our methods is much greater (upto 90%). On slopes and stepping stones the noisy and blind baselines get similar average distances and mean time to fall and both are worse than our policy. This trend is even more significant on the stepping stones terrain where all baselines barely make any progress while our methods travel upto 20m. The blind policy has no way of estimating the position of the stone and crashes as soon as it steps into the gap. For the noisy policy, the large amount of added noise makes it impossible for the student to reliably ascertain

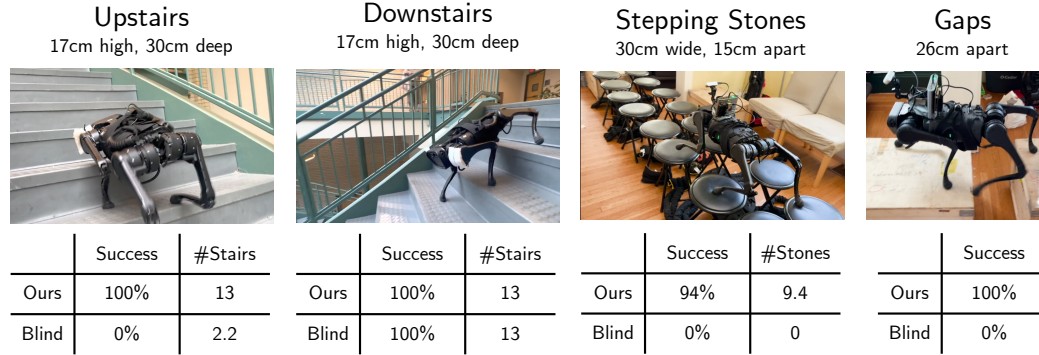

| | Upstairs 17cm high, 30cm deep | | | Downstairs 17cm high, 30cm deep | | | Stepping Stones 30cm wide, 15cm apart | | | Gaps 26cm apart |
|---|---|---|---|---|---|---|---|---|---|---|
| | Success | #Stairs | | Success | #Stairs | | Success | #Stones | | Success |
| Ours | 100% | 13 | Ours | 100% | 13 | Ours | 94% | 9.4 | Ours | 100% |
| Blind | 0% | 2.2 | Blind | 100% | 13 | Blind | 0% | 0 | Blind | 0% |

Figure 4: We show success rates and time-to-failure (TTF) for our method and the blind baseline on curbs, stairs, stepping stones and gaps. We use a separate policy for stairs which is distilled to front camera, and use a separate policy trained on stepping stones distilled to the top camera which we use for gaps and stepping stones. We observe that our method solves all the tasks perfectly except for the stepping stone task in which the robot achieves 94% success. The blind baseline fails completely on gaps and stepping stones. For upstairs, it makes some progress, but fails to complete the entire staircase even once, which is expected given the small size of the robot. The blind policy completes the downstairs task 100% success, although it learns a very high impact falling gait to solve the task. In our experiments, the robot dislocates its real right leg during the blind downstairs trials.

the location of the stones since it cannot rely on proprioception any more. We note that the blind baseline is better than the noisy one on stairs. This is because the blind baseline has learnt to use proprioception to figure out location of stairs. On the other hand, the noisy policy cannot learn to use proprioception since it is trained via supervised learning. However, the blind baseline bumps into stairs often is not very practical to run on the real robot. The noisy baseline works well in [8] possibly because of predefined foot motions which make the phase 2 learning easier. However, as noted in sec. 1, predefined motions will not work for our small robot.

**Real World Comparisons**   We compare the performance of our methods to the blind baseline in the real world. In particular we have 4 testing setups as shows in fig. 4: Upstairs, Downstairs, Gaps and Stepping stones. While we train a single phase 1 policy for all terrain, for running baselines, we obtain different phase 2 policies for stairs vs. stepping stones and gaps. Different phase 2 policies are obtained by changing the location of the camera. We use the in-built camera inside the robot for stairs and a mounted external camera for stepping stones and gaps. The in-built camera is less prone to damage but the stepping stones are gaps are not clearly visible since it is horizontal. This is done for convenience, but we also have a policy that traverses all terrain using the same mounted camera.

We see that the blind baseline is incapable of walking upstairs beyond a few steps and fails to complete the staircase even once. Although existing methods have shown stairs for blind robots, we note that our robot is relatively smaller making it a more challenging task for a blind robot. On downstairs, we observe that the blind baseline achieves 100% success, although it learns to fall on every step and stabilize leading to a very high impact gait which led to the detaching of the rear right hip of the robot during our experiments. We additionally show results in stepping stones and gaps, where the blind robot fails completely establishing the hardness of these setups and the necessity of vision to solve them. We show a 100% success on all tasks except for stepping stone on which we achieve 94% success, which is very high given the challenging setup.

**Urban Environments**   We experiment on stairs, ramps and curbs (fig. 1). The robot was successfully able to go upstairs as well as downstairs for stairs of height upto 24cm in height and 28cm as the lowest width. Since the robot has to remember terrain under its body from visual history, it sometimes misses a step, but shows impressive recovery behaviour and continues climbing or descending. The robot is able to climb curbs and obstacles as high as 26cm which is almost as high as the robot 2. This requires an emergent hip abduction movement because the small size of the robot doesn't leave any space between the body and stair for the leg to step up. This behavior emerges because of our tabula rasa approach to learning gaits without reliance on priors or datasets of natural motion.

**Gaps and Stepping Stones**    We construct an obstacle course consisting of gaps and stepping stones out of tables and stools (fig. 4). For this set of experiments we use a policy trained on stepping stones on gaps, and distilled onto the top camera instead of the front camera. The robot achieves a 100% success rate on gaps of upto 26cm from egocentric depth and 94% on difficult stepping stones. The stepping stones experiment shows that our visual policy can learn safe foothold placement behavior even without an explicit elevation map or foothold optimization objectives. The blind baseline achieves zero success rate on both tasks and falls as soon as any gap is encountered.

**Natural Environments**    We also deploy our policy on outdoor hikes and rocky terrains next to river beds (fig. 1). We see that the robot is able to successfully traverse rugged stairs covered with dirt, small pebbles and some large rocks. It also avoids stumbling over large tree roots on the hiking trail. On the beach, we see that the robot is able to successfully navigate the terrain despite several slips and unstable footholds given the nature of the terrain. We see that the robot sometimes gets stuck in the crevices and in some cases shows impressive recovery behavior as well.

## 5    Related Work

**Legged locomotion**    Legged locomotion an important problem which has been studied for decades. Several classical works use model based techniques, or define heuristic reactive controllers to achieve the task of walking  [20, 21, 22, 23, 24, 25, 26, 27, 28, 29, 30, 31, 32].This method has led to several promising results in the real world, although they still lack the generality needed to deploy them in the real world.  This has motivated work in using RL for learning to walk in simulation [15, 33, 34, 35], and then successfully deploy them in a diverse set of real world scenarios [36, 37, 38, 39, 40, 41, 36, 42].  Alternatively, a policy learned in simulation can be adapted at test-time to work well in real environments [43, 44, 45, 46, 47, 48, 49, 50, 3, 51, 52, 53]. However, most of these methods are blind, and only use proprioceptive signal to walk.

**Locomotion from Elevation Maps**    To achieve visual control of walking, classical methods decouple the perception and control aspects, assuming a perfect output from perception, such as an elevation map, and then using it for planning and control [54, 55, 56, 57, 58]. The control part can be further decoupled into searching for feasible footholds on the elevation map and then execute it with a low-level policy Chestnutt [59].  The foothold feasibility scores can either be estimated heuristically [60, 61, 10, 62, 63, 9, 64] or learned [65, 66, 67, 68, 69]. Other methods forgo explicit foothold optimization and learn traversability maps instead [70, 71, 72, 73].  Recent methods skip foothold planning and directly train a deep RL policy that takes the elevation map as input and outputs either low-level motor primitives [8, 74] or raw joint angles [18, 75, 76, 77]. Elevation maps can be noisy or incorrect and dealing with imperfect maps is a major challenge to building robust locomotion systems.  Solutions to this include incorporating uncertainty in the elevation map [54, 78, 11] and simulating errors at training time to make the walking policy robust to them [8].

**Locomotion from Egocentric Depth**    Closest to ours is the line of work that doesn't construct explicit elevation maps and predicts actions directly from depth. [53] learn a policy for obstacle avoidance from depth on flat terrain,[79] train a hierarchical policy which uses depth to traverse curved cliffs and mazes in simulation,[80] use lidar scans to show zero-shot generalization to difficult terrains.  Yu et al. [81] train a policy to step over gaps by predicting high-level actions using depth from the head and below the torso. Relatedly,  Margolis et al. [82] train a high-level policy to jump over gaps from egocentric depth using a whole body impulse controller.  In contrast, we directly predict target joint angles from egocentric depth without constructing metric elevation maps.

## 6    Discussion and Limitations

In this work, we show an end-to-end approach to walking with egocentric depth that can traverse a large variety of terrains including stairs, gaps and stepping stones. However, there can be certain instances where the robot fails because of a visual or terrain mismatch between the simulation and the real world. The only solution to this problem under the current paradigm is to engineer the situation back into simulation and retrain. This poses a fundamental limitation to this approach and in future, we would like to leverage the data collected in the real world to continue improving both the visual and the motor performance. Currently, our robot is only able to move through the environment but not interact with it meaningfully. A future direction could be to combine vision-based policies with an articulated arm [83].

**Acknowledgments**

We would like to thank Kenny Shaw and Xuxin Cheng for help with hardware. Shivam Duggal, Kenny Shaw, Xuxin Cheng, Shikhar Bahl, Zipeng Fu, Ellis Brown helped with recording videos. We also thank Alex Li for proofreading. The project was supported in part by the DARPA Machine Commonsense Program and ONR N00014-22-1-2096.

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
