# OpenReview forum: "Legged Locomotion in Challenging Terrains using Egocentric Vision"
_robot-learning.org/CoRL/2022/Conference — CoRL 2022 Oral_

### Official Review · Reviewer_YYJj · 2022-07-21

**Originality:** Very Good
**Technical Quality:** Very Good
**Clarity Of Presentation:** Fair
**Impact:** 4

**Recommendation:**

Strong Accept: I recommend accepting the paper and will argue for my recommendation even if other reviewers hold a different opinion.

**Summary:**

In this paper, the authors propose a two-stages training pipeline that enables quadruped robot to learn how to traverse uneven terrains using an onboard depth camera as inputs and without any SLAM algorithms. In the first stage of reinforcement learning, the teacher can leverage terrain elevation maps in front of the robot. Once trained, the teacher policy’s behavior is then cloned by a student policy using front mounted depth cameras in a second stage training. The authors train the policies in a diverse set of uneven terrains and deploy in the real world directly; The A1 robot demonstrates robust performance in various urban environments and on stepping stones.

**Issues:**

See my comments in the "Strengths And Weaknesses" section.

**Quality Of The Limitations Section:**

Additional details required

**Reviewer Expertise:**

5: The reviewer is absolutely certain that the evaluation is correct and very familiar with the relevant literature

**Robotics Focus:**

Sufficient demonstration on hardware

**Strengths And Weaknesses:**

Over all, it is an impressive paper with impressive results, especially on the hardware. It is the first work that I am aware of, which uses egocentric vision + pure learning to solve stepping stones. It shows the possibility of pure reinforcement learning to solve such challenging locomotion problems.


That said, here are my comments on the details of the paper:

1, Line 5, “The elevation mapping, however, is susceptible to failure and large noise artifacts, requires specialized hardware, and is biologically implausible. ” I don’t think we necessarily need to be “biologically plausible” for robots. Anyway, animals don’t use motors and have a flexible spine to support more complex locomotion behaviors.

3, Footnote on page 2, “A1 standing height is 40cm”. Should be 26 cm according to Figure 2.

4, Theorem 2.1 is the main argument on why the proposed method can work better compared with baselines (i.e. information asymmetry between teacher and student policies). However, it does not cover the case where the state spaces between two training policies (from the two stages) are different. So I don’t think it is a concrete argument.

5, It is unclear if the final policy learns the relative position of the environment, or if it learns a certain behavior that can conquer the task.

6, The teacher policy uses an elevation map in front of the robot, but not below. Has the authors tried to have a teacher policy that uses the full elevation map (i.e. even below the robot) to see if there is a behavior difference?

7, Line 91, “ This information bottleneck etc”. Reducing the number of neurons in a layer does not necessarily mean it is an information bottleneck, since these are floating point numbers. To correctly enforce an information bottleneck one needs to add the mutual information, as described in “Learning Agile Robotic Locomotion Skills by Imitating Animals”.

8, Line 105, “the ith rigid body”. Do you mean the feet?


9, Line 153. From the authors description it sounds like there is a single policy which handles or terrains. From the video the stepping stone policy seems to use an overhead camera that is looking ~45 degrees downwards, as can be told from the depth camera. However this same camera configuration does not appear in other environments such as stair climbing. This disparity in hardware configuration indicates that different policies are used. Please clarify this.


10, For all the baselines, please also report the teacher policy’s performance since this is a two stage training. Also please report the training curve and number of samples needed to obtain a policy in each stage.


**Summary Of Recommendation:**

I recommend accepting the paper conditioned that the authors revised the paper according to my suggestions.

---

> ### Author Response · Authors · 2022-08-23
> **Response to reviewer YYJj (with new experiments and real-world results) [part 1/2]**
>
> Dear Reviewer,
>
> Thanks a lot for your detailed comments. All the experiments you suggested were insightful and we are pleased to report that we finished all of them with videos here: [https://blindsupp.github.io/visual-walking/rebuttal/r4.html](https://blindsupp.github.io/visual-walking/rebuttal/r4.html)
>
> > *9, Line 153. From the authors description it sounds like there is a single policy which handles or terrains. From the video the stepping stone policy seems to use an overhead camera that is looking ~45 degrees downwards, as can be told from the depth camera. However this same camera configuration does not appear in other environments such as stair climbing. This disparity in hardware configuration indicates that different policies are used. Please clarify this.*
> - Indeed, all the settings use the same policy and we just distill them into two different camera locations. A camera location that is at a slight 45degree angle is the ideal one because it can see all terrains cleanly. But our camera mount was too fragile and was susceptible to breaking if the robot falls in outdoor challenging environments like rocks/stairs etc. So we distilled the same policy into the front camera as well which is located inside the front head of the robot, and hence, safe during falls.
> - Our policy generalizes to different camera locations through distillation.
> - However inspired by your comment, we decided to use the exact same distilled policy with a single camera location for all the experiments, and hence, designed a new 3D printed mount for the camera that is robust and works for all scenarios. We use this same camera location to run the robot on stairs, rock, curbs and stepping stones. We will include these results in the camera ready too and are very thankful to you for raising this point!
> - Please see the new videos here: [https://blindsupp.github.io/visual-walking/rebuttal/r4.html](https://blindsupp.github.io/visual-walking/rebuttal/r4.html)
>
> > *4 Theorem 2.1 is the main argument on why the proposed method can work better compared with baselines (i.e. information asymmetry between teacher and student policies). However, it does not cover the case where the state spaces between two training policies (from the two stages) are different. So I don’t think it is a concrete argument.*
> - This is a great point and true that in its current form Theorem 2.1 does not cover the case where state spaces are different between two phases. However, fortunately, we can modify the statement of the theorem to account for this. The proof in the appendix still applies with simple changes to the notation. The modified statement is
> Theorem 2.1 (modified) $\mathcal{M} = \left(\mathcal{S}, \mathcal{A}, P, R, \gamma\right)$ be an MDP with state space $\mathcal{S}$, action space $\mathcal{A}$, transition function $P:\mathcal{S}\times\mathcal{A}\rightarrow\mathcal{S}$, reward function $R:\mathcal{A}\times\mathcal{S}\rightarrow\mathbb{R}$ and discount factor $\gamma$. Let $V^1(s)$ be the value function of the phase 1 policy that is trained to be close to optimal value function $V^*(s)$, i.e., $\left|V^\ast(s) - V^1(s)\right| < \epsilon$ for all $s\in\mathcal{S}$, and $\pi^1(s)$ be the greedy phase 1 policy obtained from $V^1(s)$. Suppose the phase 2 policy operates in a different state space $\mathcal{S}'$ given by a mapping $f:\mathcal{S}\rightarrow\mathcal{S}'$ . If the phase 2 policy is close to phase 1 $\left|\pi^1(s) - \pi^2(f(s))\right| < \eta \ \forall \ s$ and $R,P$ are Lipschitz continuous, then the return of phase 2 policy is close to optimal everywhere, i.e., for all $s$,
> $$\left|V^\ast(s) - V^{\pi^2}(f(s))\right| < \frac{2\epsilon\gamma + \eta c}{1-\gamma}$$
> - The main idea is to introduce a mapping function $f$ (possibly non-deterministic) that maps from the state space of phase 1 policy to phase 2 policy. We will include this modified version along with the proof in the revised version.
>
> > *5, It is unclear if the final policy learns the relative position of the environment, or if it learns a certain behavior that can conquer the task.*
> - We believe the network implicitly learns a version of foot-hold planning. To see this, we replaced our convolutional depth backbone with a lightweight ViT and visualized the attention maps. While traversing stepping stones, a task that requires a high degree of foothold planning, we find that the backbone places high attention near the edges of the stones where the robot is about to step.
> - Video here: [https://blindsupp.github.io/visual-walking/rebuttal/r4.html](https://blindsupp.github.io/visual-walking/rebuttal/r4.html)
>
> [1/2]...continued below...

---

> > ### Author Response · Authors · 2022-08-23
> > **Response to reviewer YYJj (with new experiments and real-world results) [part 2/2]**
> >
> > > *10, For all the baselines, please also report the teacher policy’s performance since this is a two stage training. Also please report the training curve and number of samples needed to obtain a policy in each stage.*
> > - We report evaluation numbers for the phase 1 policies of Privileged and Our method below. Note that Privileged and Noisy have the same phase 1 policies. We will include the plots and tables in the revised version. This version of the Privileged baseline was trained with execution lag in phase 1.
> > - Phase 1 training takes 9.8B envsteps using 4096 parallel envs while phase 2 training takes 6M envsteps using 256 parallel envs.
> > - Training curves: https://blindsupp.github.io/visual-walking/rebuttal/r4.html](https://blindsupp.github.io/visual-walking/rebuttal/r4.html
> >
> > |                    | &#8192;&#8192; &#8192;&#8192; &#8192;&#8192;  &#8192;&#8192; &#8192;&#8192;  average  |  distance (m)  &#8192;&#8192; &#8192;&#8192; &#8192;&#8192;   |      &#8192;&#8192; &#8192;&#8192; &#8192;&#8192; mean time  |  to termination (s)    &#8192;&#8192;&#8192;&#8192;   &#8192;&#8192; &#8192;&#8192;&#8192;&#8192;    |
> > |--------------------|:----------------------:|:--------------:|:------------------------------:|:--------------:|
> > |                    | Privileged teacher   | Ours teacher | Privileged teacher           | Ours teacher |
> > | Slopes             | 55.326706            | 46.409756    |  119.2820408                  | 114.476      |
> > | Stepping Stones    | 36.003704            | 25.096626    | 70.45955556                  | 57.2976      |
> > | Stairs             | 57.842613            | 33.580025    |  118.6796                     | 83.01591837  |
> > | Discrete Obstacles | 45.22591             | 36.210846     |97.22                        | 90.97617021  |
> >
> > > *6, The teacher policy uses an elevation map in front of the robot, but not below. Has the authors tried to have a teacher policy that uses the full elevation map (i.e. even below the robot) to see if there is a behavior difference?*
> > - Yes, in initial experiments we had tried training teacher policies with full height scan map. There was no qualitative behavior difference between using a full height scan map vs. only in front of the robot. The final performance obtained in terms of rewards was also similar. We believe this is because of our recurrent policy which learns to integrate information from a history of height scans to obtain information about terrain under the robot.
> >
> > > *7, Line 91, “ This information bottleneck etc”. Reducing the number of neurons in a layer does not necessarily mean it is an information bottleneck, since these are floating point numbers. To correctly enforce an information bottleneck one needs to add the mutual information, as described in “Learning Agile Robotic Locomotion Skills by Imitating Animals”.*
> > - Thank you! Yes, you are right that a principled way to enforce an information bottleneck is indeed to add penalties as described in the linked paper. In our paper, we were inspired by other papers in this space which enforce a bottleneck by reducing the number of neurons [3,9] and it works well in practice.
> >
> > > *8, Line 105, “the ith rigid body”. Do you mean the feet?*
> > - Yes, the indices $i$ lie in the set $\mathcal{F}$ - the set of feet indices. We will simplify this to "the $i^{\textrm{th}}$ foot" as suggested.
> >
> > > *3, Footnote on page 2, “A1 standing height is 40cm”. Should be 26 cm according to Figure 2.*
> > - Thanks for pointing this out. We reported the height to the top of the robot by mistake instead of the height to its base. We will correct this to $26\textrm{cm}$ in the revised version.
> >
> > References
> >
> > [3] Kumar, Ashish, et al. "Rma: Rapid motor adaptation for legged robots." arXiv preprint arXiv:2107.04034 (2021).
> >
> > [9] Miki, Takahiro, et al. "Learning robust perceptive locomotion for quadrupedal robots in the wild." Science Robotics 7.62 (2022): eabk2822.
> >
> > [2/2]

---

### Official Review · Reviewer_bvG9 · 2022-07-28

**Originality:** Good
**Technical Quality:** Excellent
**Clarity Of Presentation:** Excellent
**Impact:** 4

**Recommendation:**

Strong Accept: I recommend accepting the paper and will argue for my recommendation even if other reviewers hold a different opinion.

**Summary:**

This paper presents a learning system for training vision-condition locomotion policies for quadruped robots that is able to traverse highly uneven terrains such as step stones, stairs, slopes, etc. The key idea is to first train a teacher policy in simulation that is conditioned on a heightmap near the robot, which is computationally cheaper, and then distill the policy into a student policy, which is conditioned on depth images, using DAGGER framework. During the training of the teacher policy, the range of the heightfield is designed to be roughly matching the fov of the depth camera, and sensor latencies are also added to facilitate student policy performance. The result method is demonstrated on a variety of indoor and outdoor uneven environments and is compared to several baseline methods. The experiments show the effectiveness of the proposed method in acquiring resilient vision-based locomotion skills for legged robots.

**Issues:**

- Prior methods that leverages teacher-student framework (e.g. [3]) learns an adaptation module for the student to predict a latent representation of the privilege information while in this work the action is used as the supervision signal. I'm wondering if this design choice would impact the algorithm performance?
- As mentioned above, the policy is trained to walk at 0.35 m/s on uneven terrains, which is well within the capability of the robot hardware. There is likely a trade-off between speed and stability in general and it can provide good insight to know what this trade-off looks like for the proposed method, e.g. at what speed does the method start to see drop in success rate.
- The teacher policy is trained with realistic heightfield range and latency. Given these two affects policy in different ways, it would be nice to have a separate ablation for them to see the contribution from each component.

**Quality Of The Limitations Section:**

Limitations are addressed clearly

**Reviewer Expertise:**

5: The reviewer is absolutely certain that the evaluation is correct and very familiar with the relevant literature

**Robotics Focus:**

Sufficient demonstration on hardware

**Strengths And Weaknesses:**

Strengths
The proposed learning method for acquiring vision-based locomotion policy is solid and effective.
The model learned from the proposed method achieved impressive results in real-world environments.
The comparison to the baseline methods show the usefulness of incorporating vision input and also learning the teacher policy in a realistic setting.

Weaknesses
Although energy and jerk-related rewards are used, the motion quality is still not at the level of methods that use reference motion. It may be interesting to incorporate ideas in a motion imitation regime to achieve more natural motion.
The movement speed seems relatively slow (0.35 m/s) and it’s not clear if this is limited by the algorithm capability.

**Summary Of Recommendation:**

The system proposed in this work enables quadruped robots to demonstrate impressive locomotion capabilities in challenging indoor and outdoor environments. The method is clearly presented and the experiments are solid. Thus I believe the work would be a good contribution to the robotics and ML community.

---

> ### Author Response · Authors · 2022-08-23
> **Response to reviewer #bvG9 (with new experiments)**
>
> Dear reviewer,
>
> Thanks a lot for insightful comments and suggestions for interesting experiments to try. We are pleased to report that we have finished all those experiments. In fact, one of your suggestions ended up increasing the performance of our method. We report our results below.
>
> > *As mentioned above, the policy is trained to walk at 0.35 m/s on uneven terrains, which is well within the capability of the robot hardware. There is likely a trade-off between speed and stability in general and it can provide good insight to know what this trade-off looks like for the proposed method, e.g. at what speed does the method start to see a drop in success rate.*
> - Thank you for this great suggestion! We fine-tune our policy to get a velocity-conditioned policy with commands in the range $[0.35\textrm{ms}^{-1}, 1.5\textrm{ms}^{-1}]$ and evaluate performance at various commanded velocities.
> |                    | average  |   distance   | travelled  (m) |    &#x7c;     | average  |  time to        |   termination(s)      |
> |--------------------|:----------------------:|:---------:|:---------:|:------:|:-----------------------------:|:---------:|:---------:|
> |                    | 0.35 m/s             | 1.0 m/s | 1.5 m/s | |0.35 m/s                    | 1.0 m/s | 1.5 m/s |
> | Slopes             | 44.3                 | 57.4    | 54.4    | |93.6                        | 56.6    | 36.5    |
> | Stepping Stones    | 15.1                 | 40.3    | 11.6  |  | 28.33                       | 39.7    | 8.4     |
> | Stairs             | 36.6                 | 48.2    | 33.2    | |75.7                        | 51.0    | 28.1    |
> | Discrete Obstacles | 34.9                 | 48.9    | 40.9 |   | 69.5                        | 47.7    | 27.9    |
>
> - The robot can walk fast on simpler terrains without issues. For speeds moderately faster than $0.35$, the robot travels a farther distance since it is moving faster. However, the performance degrades on challenging terrains like stepping stones, discrete obstacles etc. This is because higher speeds demand more precise foot placement and estimation of terrain which becomes challenging given the low frequency of depth data.
>
> > *The teacher policy is trained with realistic heightfield range and latency. Given these two affects policy in different ways, it would be nice to have a separate ablation for them to see the contribution from each component.*
> - This is a great suggestion! Turns out that doing so actually ended up increasing the performance of our method. That is, removing height-field data lag during our teacher training leads to an improved policy. So thank you very much!
> - Note we only remove lag in the incoming height-field data, we still have an execution lag for the proprioception as that is crucial to maintain the performance.
>
> > *Prior methods that leverages teacher-student framework (e.g. [3]) learns an adaptation module for the student to predict a latent representation of the privilege information while in this work the action is used as the supervision signal. I'm wondering if this design choice would impact the algorithm performance?*
> - This is a great question. Since the focus of the paper was on exploring egocentric vision for the task of walking with vision, we did not explore that design choice in our paper.
> - We did some preliminary investigation. Note that there are two diffeerent kinds of adaptation, one from proprioceptive data [3] and from vision. For the former (proprioception), the choice of action vs latent loss doesn’t matter as much. For the latter (vision), action loss seems to be critical unless you condition the vision branch on the velocity. We will investigate it more thoroughly and include the findings in the paper.

---

### Official Review · Reviewer_evpZ · 2022-08-01

**Originality:** Very Good
**Technical Quality:** Good
**Clarity Of Presentation:** Very Good
**Impact:** 4

**Recommendation:**

Strong Accept: I recommend accepting the paper and will argue for my recommendation even if other reviewers hold a different opinion.

**Summary:**

This paper proposes a new learning based controller for legged robots which uses vision to navigate difficult terrain. Particularly they propose using depth images rather than elevation maps as perceptive input. The show extensive experiments on a real robot.

**Issues:**

- Line 62 "that has access to a depth map instead..." I think should be **depth image** instead

- It's not clear to me the reason for using depth images in phase 2. Converting depth images to elevation maps is fairly easy to compute and I would expect providing an elevation map would help learning. The authors claim elevation maps are noisier, but elevation maps are just another representation of the same noisy depth data in the images so I am not convinced. Perhaps a motivating example would be a good addition.

- Following from my previous point, in my opinion the experiments do not completely justify the use of depth images. This is because the elevation maps used for training in Privileged and Noisy did not include lag which can be a huge failure point for sim-2-real problems. The authors should repeat those experiments but add lag and latency to the training of Noisy.

- There should be a more robust discussion of why depth images enable foot step placement for stepping stones or for climbing high stairs. It seem non-intuitive to me and the authors don't provide any explanation of what the network is learning or how this might be working.

**Quality Of The Limitations Section:**

Additional details required

**Reviewer Expertise:**

4: The reviewer is confident but not absolutely certain that the evaluation is correct

**Robotics Focus:**

Sufficient demonstration on hardware

**Strengths And Weaknesses:**

Strengths:
- Technically this appears to be a very strong paper. As far as I am aware, this is the first end-to-end learning controller capable to navigating stepping stones.
- The paper is very clear in the explanation of the method.

Weaknesses:
- I feel there is some discussion lacking in the paper. The authors don't really propose explanations for why their method can plan footsteps other than claim elevation maps have more noise. But I am not convinced on this point since depth images can also be very noisy.
- I don' believe the experiment properly prove depth images are superior to elevation maps because they are not a proper A to B experiment.

**Summary Of Recommendation:**

I think this paper shows significant promise and is simply lacking in discussion of the results. As such I recommend accept and have detailed my issues with the paper below. I hope the authors can expand on their paper.

---

> ### Author Response · Authors · 2022-08-23
> **Response to reviewer #evpZk (with new experiments) [part 1/2]**
>
> Dear Reviewer,
>
> Thanks a lot for your feedback and insightful questions. We realized that there might be a confusion regarding the term 'elevation map' because we accidentally overloaded it twice which we clarify below. We are also pleased to report that we finished the experiments you suggested.
>
> We hope that these answers (+ new experiments) address your concerns, and if so, we request the reviewer to consider updating the score. Otherwise, please let us know any further concerns that remain.
>
> ### Clarification about elevation mapping vs. our method (with appropriate revisions)
> > *It's not clear to me the reason for using depth images in phase 2. Converting depth images to elevation maps is fairly easy to compute and I would expect providing an elevation map would help learning. The authors claim elevation maps are noisier, but elevation maps are just another representation of the same noisy depth data in the images so I am not convinced. Perhaps a motivating example would be a good addition.*
> - We believe there is a confusion here due to the usage of the term 'elevation map' because in the paper, we accidentally overloaded the term 'elevation map' to refer to two completely different things.  (1) In the intro, we use 'elevation maps' to mean the sensor-fusion variant used in the literature [8, 9, 10] which we describe below in more detail. (2) In the methods section, we erroneously refer to the height measurements $\mathbf{m}_t$ we feed to our phase 1 policy as 'elevation maps'. These should technically be called 'scandots', because they are directly observed by the agent (kind of like a coarse depth image) and are not a result of TSDF fusion of multiple observations. Hence, they do not cover all the surface around/under the robot unlike traditional elevation maps.
> - **Elevation map with sensor fusion** (prior work): Most prior locomotion techniques rely on the elevation map of the terrain around and under the robot [8,9,10] to plan foot steps and joint angles. This is done by fusing information from multiple depth images (collected over time). This fusion of depth images into a single elevation map requires the relative pose between cameras at different times. Hence, tracking is required in the real world to obtain this relative pose using visual or inertial odometry. This is challenging because of noise introduced in sensing and odometry, and hence, previous methods add different kinds of structured noise at training time to account for the noise due to pose estimation drift [84,75,85].
> - **scan dots (phase 1) and depth (phase 2)** (our setup):  Scan dot values are computed in simulation from the height map of the terrain. Although an approximate version of this information can be computed in the real world, the height of the terrain behind an occlusion, and very extreme height values like the holes in stepping stones cannot be computed directly from the depth and imu information. In phase two, instead of depth, we could very well have used the transformed version of depth to generate approximate scan dots, but to keep the pipeline simple, we directly use raw depth sensor information.
> - Prior vs Ours: There is no pose estimation involved in directly observing depth or scan dots in contrast to elevation map obtained through fusion. Hence, your comment about not needing tracking for our approach is correct. We simply input all raw data (depth, proprioception) into the policy and let the network extract useful information, bypassing the odometry problem entirely. Hence, to show that our approach is outperforms elevation mapping, we compare against the elevation mapping baseline (called Noisy baseline). It takes the whole elevation map around/under the robot as input with added noise as in [9] . This baseline is equivalent to [9] and we use their proposed noise model that closely approximates the noise observed in the real world.
> - We apologize for this confusion and will fix this term overloading in the camera ready version by referring to $\mathbf{m}_t$ as scandots.
>
> [1/2]...continued below...

---

> > ### Author Response · Authors · 2022-08-23
> > **Response to reviewer #evpZk (with new experiments) [part 2/2]**
> >
> > ### New experiment: Noisy baseline with execution lag
> > > *Following from my previous point, in my opinion the experiments do not completely justify the use of depth images [...] The authors should repeat those experiments but add lag and latency to the training of Noisy.*
> > - We hope our previous answer has clarified the confusion of the term elevation map in baseline/introduction vs. the one in our phase 1 (which we will call scandots).
> > - However, that being said, reviewer’s point about lag is a great suggestion. As per the suggestion, we repeated the Noisy experiments but added lag and latency. With this change, the performance of Noisy slightly increases but is still outperformed by our approach. We will replace the Noisy baseline in paper with this one. Exact numbers are here:
> >
> > |                    | &#8192;&#8192; ‎average | distance(m) | &#65372; | &#8192;mean time to  |  termination (s)   |
> > |--------------------|:----------------------:|:------:|:--:|:------------------------------:|:------:|
> > |                    | Noisy + lag  | Ours | | Noisy + lag             | Ours |
> > | Slopes             | 18.3                   | **31.5** | |98.7                           | 89.2 |
> > | Stepping Stones    | 1.6               | **18.1** | |3.7                           | 46.3 |
> > | Stairs             | 36.9                  | **23.8** | |110.6                           | 60.5 |
> > | Discrete Obstacles | 17.7                   | **30** |  | 94.3                       | 80.6 |
> >
> > ### New experiment: Visualization of depth backbone
> > > *There should be a more robust discussion of why depth images enable foot step placement for stepping stones or for climbing high stairs. It seems non-intuitive to me and the authors don't provide any explanation of what the network is learning or how this might be working.*
> > - We believe the network implicitly learns a version of foot-hold planning. As per your suggestion, to substantiate this claim, we replaced our convolutional depth backbone with a lightweight ViT and visualized the attention maps. While traversing stepping stones, a task that requires a high degree of foothold planning, we find that the backbone places high attention near the edges of the stones where the robot is about to step.
> > - Please see the video here: [https://blindsupp.github.io/visual-walking/rebuttal/r2.html](https://blindsupp.github.io/visual-walking/rebuttal/r2.html)
> >
> > References
> >
> > [8] F. Jenelten, T. Miki, A. E. Vijayan, M. Bjelonic and M. Hutter, "Perceptive Locomotion in Rough Terrain – Online Foothold Optimization," in IEEE Robotics and Automation Letters, vol. 5, no. 4, pp. 5370-5376, Oct. 2020, doi: 10.1109/LRA.2020.3007427.
> >
> > [9] Miki, Takahiro, et al. "Learning robust perceptive locomotion for quadrupedal robots in the wild." Science Robotics 7.62 (2022): eabk2822.
> >
> > [10] Kim, Donghyun, et al. "Vision aided dynamic exploration of unstructured terrain with a small-scale quadruped robot." 2020 IEEE International Conference on Robotics and Automation (ICRA). IEEE, 2020.
> >
> > [75] Fankhauser, Péter, et al. "Robot-centric elevation mapping with uncertainty estimates." Mobile Service Robotics. 2014. 43k3-440.
> >
> > [84] Miki, Takahiro, et al. "Elevation Mapping for Locomotion and Navigation using GPU." arXiv preprint arXiv:2204.12876 (2022).
> >
> > [85] Pan, Yiyuan, et al. "GEM: online globally consistent dense elevation mapping for unstructured terrain." IEEE Transactions on Instrumentation and Measurement 70 (2020): 1-13.
> >
> > [2/2]

---

> > > ### Comment · Reviewer_evpZ · 2022-08-23
> > > **Response**
> > >
> > > Thank you for your clear response. Provided the explanation of scandots vs elevation map and the additional experiment of Noisy with lag are added to a revised version of the paper (On Open Review I currently can only see the original submission), then I would be happy to revise my recommendation to Strong Accept.

---

> > > > ### Author Response · Authors · 2022-08-23
> > > > **Response: updating the paper**
> > > >
> > > > Dear Reviewer,
> > > >
> > > > We are grateful to you for the quick and promising reply. We wanted to update the paper on openreview immediately but the email from CoRL PCs said that all the updated papers will become public right away to everyone. Note currently only the abstract and the review discussion is public to everyone but the paper is hidden. Making the full paper public is slightly uncomfortable for us because it's anonymous. However, we promise you that we will update the paper with all the new experiments you (and other reviewers) suggested and they make the paper stronger.
> > > >
> > > > Would you be able to revise your recommendation based on this context? If not, please let us know, then we will ask PCs if there is a special mechanism to share the updated paper with you or will release it publicly to everyone.
> > > >
> > > > Thank you very much once again. We look forward to your reply.

---

> > > > > ### Author Response · Authors · 2022-08-27
> > > > > **Request for follow up (deadline today)**
> > > > >
> > > > > Dear Reviewer,
> > > > >
> > > > > We gently wanted to bump this thread again as today is the last day of discussion and you mentioned *revising the recommendation to Strong Accept*. We got this email from PCs saying:
> > > > >
> > > > > > “**Q: Can I edit my original submission PDF?** A: Not during the rebuttal phase. If the paper is accepted, you will have to upload a final camera-ready version of the paper. Any uploads during the rebuttal phase will NOT change the original or final paper. ”
> > > > >
> > > > > We promise that we will update the paper with all the new experiments you suggested because they strictly increase the quality of our paper. If there are any other concerns or comments, we are happy to address them.
> > > > >
> > > > > Thank you very much for your time!

---

### Official Review · Reviewer_9Sia · 2022-08-03

**Originality:** Very Good
**Technical Quality:** Very Good
**Clarity Of Presentation:** Very Good
**Impact:** 4

**Recommendation:**

Strong Accept: I recommend accepting the paper and will argue for my recommendation even if other reviewers hold a different opinion.

**Summary:**

The paper proposes an end-to-end training for legged locomotion in significantly uneven terrains. The end policy is a recurrent network that uses depth camera images and trained in simulation to overcome different types of terrains that requires vision. It is trained in two phases, first in simulation  using RL then using supervised training using real depth images.

**Issues:**

No issues, but some questions I mentioned in 'strengths and weaknesses'

**Quality Of The Limitations Section:**

Limitations are addressed clearly

**Reviewer Expertise:**

4: The reviewer is confident but not absolutely certain that the evaluation is correct

**Robotics Focus:**

Sufficient demonstration on hardware

**Strengths And Weaknesses:**

The paper shows the robot walking over very difficult terrains for a small robot. The results are very impressive. The proposed method is combination of a previously known methods but it is the first time I have seen it applied to the legged locomotion, so I would consider it novel.

The method is very well explained and it is compared to some baselines such as blind policy or privileged teacher. One concern I have is about the privileged teacher performing very poorly. The result was very suprising and I suspect it is due to the lack of latency in training. I think it would be a more fair comparison if the authors kept the same dynamics but just used privileged teacher for vision. The results might be more competitive to the proposed method.

I think the impact of the paper would be higher if the authors could show the robot walking slightly faster or analyze / explain at what point and why it fails.

**Summary Of Recommendation:**

The paper is novel within the context of legged locomotion. The results are impressive and there is significant amount of hardware results.

---

> ### Author Response · Authors · 2022-08-22
> **Response to reviewer #9Sia (with new experiments)**
>
>
> Dear Reviewer,
>
> Thank you for the insightful and positive feedback! We provide clarifications for your concerns below.
>
> > *I think the impact of the paper would be higher if the authors could show the robot walking slightly faster or analyze / explain at what point and why it fails.*
> - Thank you for this great suggestion! We fine-tune our policy to get a velocity-conditioned policy with commands in the range $[0.35\textrm{ms}^{-1}, 1.5\textrm{ms}^{-1}]$ and evaluate performance at various commanded velocities.
>
> |                    | average  |   distance   | travelled  (m) |    &#x7c;     | average  |  time to        |   termination(s)      |
> |--------------------|:----------------------:|:---------:|:---------:|:------:|:-----------------------------:|:---------:|:---------:|
> |                    | 0.35 m/s             | 1.0 m/s | 1.5 m/s | |0.35 m/s                    | 1.0 m/s | 1.5 m/s |
> | Slopes             | 44.3                 | 57.4    | 54.4    | |93.6                        | 56.6    | 36.5    |
> | Stepping Stones    | 15.1                 | 40.3    | 11.6  |  | 28.33                       | 39.7    | 8.4     |
> | Stairs             | 36.6                 | 48.2    | 33.2    | |75.7                        | 51.0    | 28.1    |
> | Discrete Obstacles | 34.9                 | 48.9    | 40.9 |   | 69.5                        | 47.7    | 27.9    |
> - The robot can walk fast on simpler terrains without issues. For speeds larger than $0.35$, the robot initially travels a farther distance since it is moving faster. However, the performance degrades on challenging terrains like stepping stones, discrete obstacles etc. This is because higher speeds demand more precise foot placement and estimation of terrain which becomes challenging given the low frequency of depth data.
>
> > *One concern I have is about the privileged teacher performing very poorly. The result was very suprising and I suspect it is due to the lack of latency in training.*
> - Yes, you are right. This experiment was only meant to provide an empirical example to verify Theorem 2.1 which states that if the phase 2 student policy can closely approximate the actions of phase 1 teaching, its performance returns will be tightly bounded. Hence, through this example, we showed that when the student can not approximate teacher well, its performance is much worse (consistent with the theorem).
> - Indeed, adding execution lag to the privileged baseline improves performance. This is also consistent with our theorem 2.1 since adding execution lag reduces information asymmetry between phase 1 and phase 2. We will add this ablation in the paper. Thank you again for the suggestion.

---

### Meta-Review · Area_Chair_roaD · 2022-08-10

**Recommendation:** Accept (Oral)
**Confidence:** 5

**Metareview:**

The paper proposes a learning-based approach for achieving legged locomotion in challenging terrains using onboard sensors.

**Strengths**
- The proposed learning method seems to be novel and effective.
- The paper shows an impressive demo of stepping stones on hardware.
- The paper is well written and easy to follow.

**Weaknesses**
- Reviewers requested a few technical details of the work for better reproducibility. Please respond to individual comments.

**Post-rebuttal updates**
N/A (in a positive way)

**Best Paper Nomination:**

Yes

---

> ### Author Response · Authors · 2022-08-22
> **Response to Area Chair #roaD (with new results in sim and real)**
>
> Dear AC,
>
> We thank you and the reviewers for the insightful and positive responses to our real-world results on challenging terrain like high stairs, curbs in the wild and stepping stones. We have clarified all the questions the reviewers had about our method below. We are also pleased to report that we finished all the experiments reviewers suggested and have reported them below both in terms of quantitative numbers and qualitative result videos in the real world.
>
> - Here are the videos of new stronger real-world results as per suggestions from Reviewer # YYJj and Reviewer # evpZk: [https://blindsupp.github.io/visual-walking/rebuttal/r4.html](https://blindsupp.github.io/visual-walking/rebuttal/r4.html)
> - Note that these contain additional results on rocks, stairs, curbs and stepping stones using the exact same policy and camera mount. We also report visualization of agent’s attention inside the deep network which show emergent footsteps.